# Factors Influencing Breast Milk Antibody Titers during the Coronavirus Disease 2019 Pandemic: An Observational Study

**DOI:** 10.3390/nu16142320

**Published:** 2024-07-18

**Authors:** Christoph Hochmayr, Ira Winkler, Marlene Hammerl, Alexander Höller, Eva Huber, Martina Urbanek, Ursula Kiechl-Kohlendorfer, Elke Griesmaier, Anna Posod

**Affiliations:** 1Department of Pediatrics II (Neonatology), Medical University of Innsbruck, Anichstraße 35, 6020 Innsbruck, Austria; 2Division for Nutrition and Dietetics, University Hospital Innsbruck, 6020 Innsbruck, Austria; 3Institute of Public Health, Medical Decision Making and Health Technology Assessment, Department of Public Health, Health Services Research and Health Technology Assessment, UMIT TIROL—University for Health Sciences and Technology, 6060 Hall in Tirol, Austria

**Keywords:** SARS-CoV-2, COVID-19, breast milk, breastfeeding, anti-S1RBD immunoglobulins, immunonutrition

## Abstract

The COVID-19 pandemic has highlighted the role of breastfeeding in providing passive immunity to infants via specific anti-SARS-CoV-2 antibodies in breast milk. We aimed to quantify these antibodies across different lactation stages and identify influencing factors. This prospective study involved mother–child dyads from Innsbruck University Hospital, Austria, with a positive maternal SARS-CoV-2 test during pregnancy or peripartum between 2020 and 2023. We collected breast milk samples at various lactation stages and analyzed anti-Spike S1 receptor-binding domain (S1RBD) immunoglobulins (Ig). Maternal and neonatal data were obtained from interviews and medical records. This study included 140 mothers and 144 neonates. Anti-S1RBD-IgA (72.0%), -IgG (86.0%), and -IgM (41.7%) were highly present in colostrum and decreased as milk matured. Mothers with natural infection and vaccination exhibited higher anti-S1RBD-IgA and -IgG titers in all milk stages. Mothers with moderate to severe infections had higher concentrations of anti-S1RBD-IgA and -IgG in transitional milk and higher anti-S1RBD-IgA and -IgM in mature milk compared to those with mild or asymptomatic infections. Variations in antibody responses were also observed with preterm birth and across different virus waves. This study demonstrates the dynamic nature of breast milk Ig and underscores the importance of breastfeeding during a pandemic.

## 1. Introduction

In the wake of the unprecedented global health crisis triggered by the COVID-19 pandemic, the significance of breastfeeding and its potential implications on infant health have garnered increased attention. In the very beginning of the pandemic, newborns were separated from SARS-CoV-2-positive mothers, and breastfeeding was discouraged due to fear of infection [1,2]. However, soon, studies reported significant levels of specific anti-SARS-CoV-2 antibodies in breast milk samples from COVID-19-recovered donors [1,3,4,5,6] and vaccinated breastfeeding mothers [5,7,8], indicating the transfer of passive immunity. Additional in vitro studies demonstrated the neutralizing capacity of SARS-CoV-2-specific antibodies in human breast milk [4,9,10].

Ever since, the World Health Organization (WHO), the U.S. Centers for Disease Control and Prevention (CDC), and other national healthcare systems have advocated for breastfeeding with active SARS-CoV-2 infection, as the benefits of human milk seem to far exceed the minimal risk of potential SARS-CoV-2 transmission [11,12].

Studies show that the mechanism for acquiring immunity influences breast milk responses in terms of titer development and predominant immunoglobulin (Ig) class; natural infection with SARS-CoV-2 induces a rapid rise in Spike-reactive IgA [3,13], whereas vaccination mainly induces an IgG response [5,14]. However, little is known about the effect of milk maturation and other factors on anti-SARS-CoV-2 Ig levels and composition in breast milk [5].

The present study aimed to quantify breast milk Ig specifically directed against the Spike S1 subunit of the SARS-CoV-2 receptor-binding domain (S1RBD) in all stages of lactation throughout the course of the pandemic and to assess factors that influence concentrations of anti-S1RBD Ig in human milk.

## 2. Materials and Methods

### 2.1. Study Design and Population

Mother–child dyads included in the present analysis were participants of the prospective study “Evaluation of potential effects of a maternal SARS-CoV-2 infection on the newborn” conducted at the Department of Pediatrics II (Neonatology) at Innsbruck University Hospital, Austria. We invited all women who gave birth at Innsbruck University Hospital between July 2020 and January 2023 and had a positive nasopharyngeal swab test result for SARS-CoV-2 during pregnancy or peripartum to participate in this study.

We employed convenience sampling and determined our sample size based on resource constraints.

This study was conducted according to the guidelines of the Declaration of Helsinki and was approved by the Institutional Review Board of the Medical University of Innsbruck, Austria (approval number 1084/2020, 11 May 2020), and all participants gave written informed consent for participation.

### 2.2. Maternal and Neonatal Data Collection

Data on maternal past medical history, first positive test results for SARS-CoV-2, and disease course and symptom severity, as well as vaccination status, were collected by means of personal interviews and medical records. Maternal SARS-CoV-2 infection was categorized as asymptomatic or pre-symptomatic disease if the mother had a positive SARS-CoV-2 test without symptoms. Mild disease was defined as flu-like symptoms, such as fever, cough, myalgia, and anosmia. Moderate disease was defined as lower respiratory tract disease with dyspnea, pneumonia, refractory fever, and abnormal blood gas results while maintaining an oxygen saturation above 93%. Severe disease was defined as a respiratory rate greater than 30 breaths per minute and the need for oxygen supplementation. Critical disease was defined as multi-organ failure, respiratory failure requiring high-flow nasal cannula therapy, or mechanical ventilation [15].

Virus waves were categorized according to the timing of infection. Data on the prevailing SARS-CoV-2 variant at any given time point were obtained from the website of the Global Initiative on Sharing all Influenza Data (GISAID) [16] and the regular local GISAID reports, which are published on the homepage of the Austrian Agency for Health and Food Safety GmBH, Vienna, Austria (AGES) [17].

All relevant perinatal and neonatal data were collected from medical records.

### 2.3. Sample Collection and Processing

#### 2.3.1. Neonatal Samples

After obtaining informed consent, every newborn was swab-tested for SARS-CoV-2. Specimens were analyzed at the ISO-certified Central Institute of Clinical and Chemical Laboratory Diagnostics at Innsbruck University Hospital, Austria.

Whole-blood anti-S1RBD-IgG and -IgM were quantified in dried blood spot (DBS) samples obtained from umbilical cord blood and/or neonatal venous blood collected at 48 h of life. DBS samples from patients without SARS-CoV-2 infection or vaccination were used as negative controls. Samples exhibiting anti-S1RBD-IgG concentrations greater than 30 U/mL and anti-S1RBD-IgM concentrations greater than 100 U/mL were considered positive. Methodological details are provided in the Appendix A.

#### 2.3.2. Breast Milk Samples

Breast milk samples were self-collected by participants into sterile 1.8 mL cryotubes (CryoPure, Sarstedt, Nümbrecht, Germany) at different time points, as follows: (i) colostrum (in the first 3 days after delivery), (ii) transitional milk (4 to 14 days after delivery), and (iii) mature milk (>14 days after delivery). Samples were centrifuged three times at 4100 rpm/4 °C for 15 min to obtain fat-free, acellular whey milk and stored at −20 °C until further analysis.

Pre-diluted samples were plated in duplicate on a 96-well plate coated with S1RBD protein and incubated for 1 h (IEQ-CoVS1RBD-IgA, -IgM and -IgG, Raybiotech, GA, USA). For background subtraction, samples were further added to a human albumin-coated plate (IgA, IgM). After washing, plates were incubated with biotinylated anti-human IgA, IgM, and IgG and, subsequently, with HRP-Streptavidin concentrate. 3,3′,5,5′-tetramethylbenzidine (TMB) substrate solution was added, and the reaction was stopped with sulphuric acid. Plates were read at 450 nm (Hidex Sense, HVD Lifesciences, Vienna, Austria). An inactivated serum sample was used to create a standard curve to calculate Ig concentrations.

To determine the detection thresholds (DT), five breast milk samples from healthy donors without prior SARS-CoV-2 infection or vaccination were included as negative controls in every run. DT was calculated as the mean of negative controls + 3× SD (DT IgA: 7.52 U/mL; DT IgG: 0.72 U/mL; DT IgM: 1.32 U/mL).

### 2.4. Statistical Analysis

A statistical analysis was performed using SPSS version 29.0 for Windows (IBM Corporation, Armonk, NY, USA) and GraphPad Prism version 10.1.2 for Windows (GraphPad Software, Boston, MA, USA). For all Ig titers, values below the respective lower limits of detection (LOD) were censored as LOD/√(2). Data distribution was evaluated by means of a histogram analysis as well as the Shapiro–Wilk test. For the analysis of non-normally distributed data, the Mann–Whitney U test was applied. If more than two groups were compared at a time, overall differences between groups were detected with the Kruskal–Wallis test. A post hoc analysis was conducted by means of the Mann–Whitney U test with Bonferroni correction for multiple comparisons. Associations between two variables were assessed by means of Spearman’s rank correlation coefficient. The results were regarded as statistically significant when *p* < 0.05 or at the respective adjusted level of significance if the Bonferroni correction was applied.

## 3. Results

A total number of 140 mothers with 144 neonates (four twin births) participated in this study. Most of the mothers were infected during pregnancy (70.0%), while 30.0% had peripartum active COVID-19. Two-thirds (67.1%) acquired SARS-CoV-2 immunity by natural infection only; the rest were additionally vaccinated. In all participating mothers, natural infection was the last antigen contact. Of the included newborns, 34 (23.6%) were admitted to the neonatal intensive care unit (NICU) due to prematurity or common neonatal disease entities. Three neonates (2.1%) tested positive in the nasal swab. None of them developed COVID-19-like symptoms or had to be admitted to the NICU. The median DBS anti-S1RBD-IgG concentrations in all participating neonates were 90.8 U/mL, while anti-S1RBD-IgM was not detectable in any of the analyzed samples. Table 1 provides detailed information on participants’ characteristics.

In breast milk, anti-S1RBD-IgA was detectable in 72.0%, -IgG was detectable in 86.0%, and -IgM was detectable in 41.7% of colostrum samples. In transitional milk samples, anti-S1RBD-IgA was detectable in 46.3%, -IgG was detectable in 53.7%, and -IgM was detectable in 33.7%. Mature milk samples were collected at a median of 16 days after delivery (25th percentile: 15 days, 75th percentile: 22 days). In mature milk samples, anti-S1RBD-IgA was detectable in 45.7%, -IgG was detectable in 50.7%, and -IgM was detectable in 23.0%. Detailed information on sample availability is provided in Appendix A. The most abundant Ig class in colostrum was anti-S1RBD-IgA, with a median concentration of 38.5 U/mL, followed by anti-S1RBD-IgG, with a median concentration of 22.2 U/mL. With milk maturation, the concentrations of anti-S1RBD Ig decreased, with median IgA concentrations falling below the detection threshold, while median IgG concentrations remained above the detection threshold. The median concentrations of anti-S1RBD-IgM were below the detection threshold in all types of breast milk samples. Details are provided in Figure 1.

Anti-S1RBD-IgA (r = 0.366, *p* = 0.002) and -IgG (r = 0.494, *p* < 0.001) concentrations in colostrum were significantly correlated with DBS anti-S1RBD-IgG concentrations in neonatal blood samples, while anti-S1RBD-IgM in colostrum and all Ig classes in other breast milk sample types did not correlate (all *p* > 0.05). We also did not observe positive or negative correlations between breast milk anti-S1RBD Ig concentrations and the interval from the last maternal antigen contact to the time of sample collection (all *p* > 0.05). However, we found significantly higher anti-S1RBD-IgA and -IgG concentrations in mature milk samples of mothers with active peripartum SARS-CoV-2 infection in comparison to those with infection during pregnancy. The concentrations of anti-S1RBD-IgA and -IgG were significantly higher in colostrum, transitional milk, and mature milk samples of mothers who had received at least one dose of mRNA vaccination in addition to natural infection, whereas anti-S1RBD-IgM concentrations were significantly lower. Other types of vaccination were not recorded in our study cohort. Mothers with moderate to severe infection had significantly higher anti-S1RBD-IgA and -IgG concentrations in transitional milk and significantly higher anti-S1RBD-IgA and -IgM concentrations in mature milk than mothers with no to mild COVID-19 symptoms, while anti-S1RBD Ig concentrations in colostrum did not significantly differ between the two groups. We also observed significantly higher anti-S1RBD-IgM concentrations in mature breast milk samples of mother–child dyads with preterm birth in comparison to term birth. No significant differences in breast milk anti-S1RBD-Ig concentrations were detected in relation to advanced maternal age (≥35 years). Detailed information is provided in Table 2.

With regard to virus waves, we detected significant overall differences in anti-S1RBD-IgG concentrations in mature milk and in anti-S1RBD-IgM concentrations in colostrum, transitional milk, and mature milk. A post hoc analysis with pairwise comparisons revealed significant Ig- and sample type-specific differences between the original wave and the Omicron wave, as well as the Delta wave and the Omicron wave. Detailed results are presented in Table 3 and Figure 2.

## 4. Discussion

Breast milk is the physiological form of infant nutrition, widely recognized for its numerous benefits, including its role as a source of immunological factors that protect infants from infectious agents [18,19]. During the COVID-19 pandemic, numerous studies have demonstrated the potential protective effect of passive immunity conveyed through Ig in breast milk [4,9,10]. However, the factors that influence immunoglobulin titers and the role of milk maturation remain incompletely understood [19].

Our study uniquely elucidates the dynamics of anti-SARS-CoV-2 antibodies in breast milk across different lactation stages. By analyzing breast milk Ig in a large cohort of mothers with diverse COVID-19 infection timelines, disease severities, and vaccination statuses throughout the pandemic, we provide comprehensive insights into breast milk immunity.

Specifically, we measured anti-S1RBD-Ig of the classes IgA, IgG, and IgM in colostrum, transitional, and mature milk, elucidating the influence of various factors on their concentrations. The most abundant Ig in the analyzed samples was anti-S1RBD-IgA, which is consistent with previous research [1,6,20]. Notably, anti-S1RBD-IgA levels were higher in colostrum than in mature milk, which also accords with prior findings in the field [19]. A high detection rate of IgG in colostrum (86.0%) corroborates earlier studies on SARS-CoV-2 antibodies in breast milk [4]. In transitional and mature milk, median anti-S1RBD-IgG concentrations remained above the detection threshold, while anti-S1RBD-IgA concentrations fell below the detection threshold. This may be attributed to the proportion of vaccinated mothers in our cohort, as vaccination is known to enhance the IgG response in breast milk [21]. For anti-S1RBD-IgM, a decline in detection rates was observed with milk maturation (41.7% vs. 33.7% vs. 23.0%), corroborating previous findings that IgM diminishes in significance during lactation [22,23]. Unlike IgG, IgM secretion into breast milk is unaffected by vaccination [10,24].

Our finding that the time since the last antigen exposure does not affect breast milk anti-S1RBD-Ig levels aligns with Szczygiol et al., who found no correlation between serum anti-SARS-CoV-2 IgA and IgG levels and the time since a positive antigen test [6]. This may also apply to breast milk, given the established correlation between maternal serum and breast milk antibodies [24,25]. Although maternal serum samples were unavailable, we found significant correlations between colostrum anti-S1RBD-IgA and -IgG and DBS anti-S1RBD-IgG in neonatal blood. The undetectable DBS anti-S1RBD-IgM suggests that these IgG antibodies are maternally derived via transplacental transfer.

In alignment with previous studies, we found higher concentrations of anti-S1RBD-IgA and -IgG in vaccinated mothers’ milk, indicating that vaccination amplifies the immune response initially triggered by natural infection [21,26]. Ramirez et al. observed robust anti-S1RBD-IgG and -IgA levels in breast milk post-mRNA vaccination, highlighting enhanced antibody titers compared to natural infection alone [26]. Our study’s results are further supported by Selma-Royo et al., who showed that breast milk anti-S1RBD-IgA and -IgG induced by mRNA-based vaccines lead to higher levels of anti-SARS-CoV-2-IgA and -IgG in breast milk compared to natural infection [21]. The lower concentrations of anti-S1RBD-IgM following combined vaccination and natural infection, compared to natural infection alone, are in accordance with previous studies showing that IgM levels in human milk do not increase with additional vaccination [10,24].

Our study indicates that the severity of maternal SARS-CoV-2 infection influences the concentration of specific anti-S1RBD-Ig in breast milk. Specifically, we observed that mothers with moderate to severe infections had significantly higher concentrations of anti-S1RBD-IgA and -IgG in transitional milk and significantly higher concentrations of anti-S1RBD-IgA and -IgM in mature milk than mothers with no to mild COVID-19 symptoms. This finding underscores the dynamic nature of the maternal immune response and its capacity to modulate breast milk composition in response to infection. It aligns with the previous literature suggesting that the severity of COVID-19 disease influences antibody titers [5]. However, the literature on this topic is conflicting, and while some studies report higher antibody levels in symptomatic individuals, other studies could not demonstrate an association between disease severity and breast milk antibody titers [6,13,27]. Comprehensive studies are required to draw definitive conclusions.

Interestingly, we detected higher mature milk anti-S1RBD-IgM titers after preterm birth. This finding accords with data indicating that preterm milk is a superior source of anti-inflammatory and anti-infective factors [28,29]. However, most research on the influence of gestational age on breast milk Ig has focused on IgA, and data on the impact of preterm birth on titers of other Ig classes are sparse. Of note, higher concentrations of IgA and IgG have been observed in preterm milk [19,30]. For IgM, earlier studies report similar concentrations in preterm and term breast milk [30,31]. However, these studies examined total Ig concentrations in breast milk in general and were not virus-specific. The impact of gestational age on breast milk responses to specific pathogens, including SARS-CoV-2, necessitates further targeted investigation.

To date, few studies have explored the impact of different virus waves on Ig concentrations in breast milk, mainly focusing on the neutralizing capacity against different virus variants following mRNA vaccination [7,25]. We are the first to report virus wave-specific anti-S1RBD differential responses depending on the type of breast milk sample and Ig class.

While our study provides valuable insights, several limitations should be acknowledged to guide future research.

First, although we did not explore potential confounding factors, such as maternal diet, stress levels, or supplementary infant feedings, our findings lay a strong foundation for subsequent studies to investigate these important aspects of infant nutrition and health.

Second, we inferred virus variants based on the timing of infection, which limits our ability to distinguish between the influences of the virus variants themselves and the public health responses during different phases of the pandemic. Including viral genome sequencing in future studies could provide more precise identification of virus variants.

Additionally, although we did not collect maternal serum samples to correlate breast milk antibody levels with maternal systemic immune activation, our findings underscore the substantial presence of antibodies in breast milk, indicating a robust adaptive maternal immune response.

Furthermore, our study was conducted at a single hospital in Austria, which might limit the generalizability of the findings. However, this controlled setting allows for a detailed and focused analysis, paving the way for multicenter studies in diverse populations. Although we did not collect data on infant outcomes or protection from COVID-19, the high levels of antibodies detected in breast milk are promising indicators of potential infant protection.

Lastly, while our focus was on breast milk antibodies, future research should explore other components of breast milk that contribute to infant immunity and assess the functional capacity of these antibodies, such as their neutralizing ability.

The major strengths of our study include its large sample size, the systematic assessment of breast milk samples at all stages of early lactation, and the comprehensive observation period spanning the entirety of the pandemic.

## 5. Conclusions

In conclusion, our study offers relevant insights into the dynamics of anti-SARS-CoV-2 Ig in breast milk, emphasizing the impact of maternal vaccination status and infection severity, as well as preterm birth on antibody titers. It underscores the crucial role of breastfeeding in providing tailored immunological support to infants, particularly during a pandemic. However, further research is necessary to understand the extent of protection breast milk offers against COVID-19 and the potential effects of different lactation phases. Future studies should investigate the specific influences of virus variants and maternal factors to further elucidate the complex immunological landscape of breast milk and explore the multifaceted benefits of breastfeeding in the context of maternal and infant health.

## Figures and Tables

**Figure 1 nutrients-16-02320-f001:**
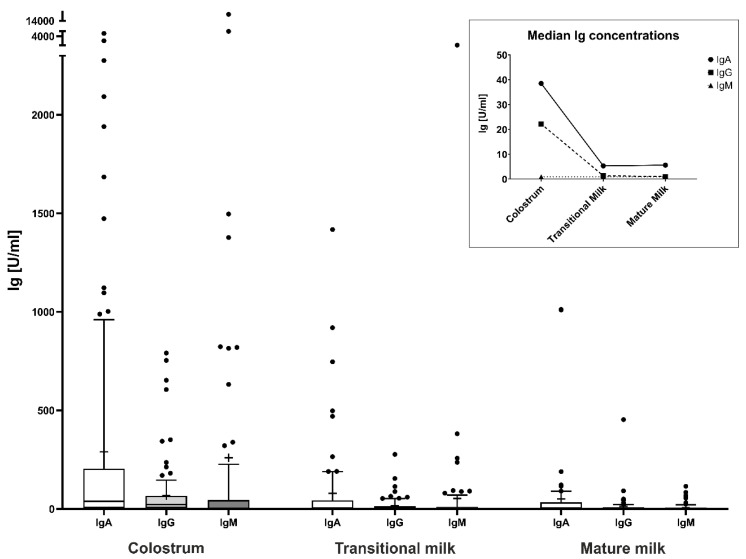
Concentrations of anti-SARS-CoV-2-S1RBD immunoglobulins (Ig) in different types of breast milk. This figure shows boxplots of anti-S1RBD-IgA, -IgG, and -IgM concentrations in various types of breast milk, including colostrum, transitional milk, and mature milk. Ig concentrations are plotted on the *y*-axis in U/mL. Center lines in boxes represent medians, box edges mark 1st and 3rd quartiles, and whiskers indicate 10th and 90th percentiles. Individual points indicate outliers. Crosses indicate mean Ig concentrations. The data in the box highlight the dynamic changes in breast milk Ig levels over the lactation period, with the highest median concentrations observed in colostrum and gradually decreasing median concentrations in transitional and mature milk. *Ig*, immunoglobulin; *SARS-CoV-2*, severe acute respiratory syndrome coronavirus 2; *S1RBD*, SARS-CoV-2 Spike protein S1 receptor-binding domain.

**Figure 2 nutrients-16-02320-f002:**
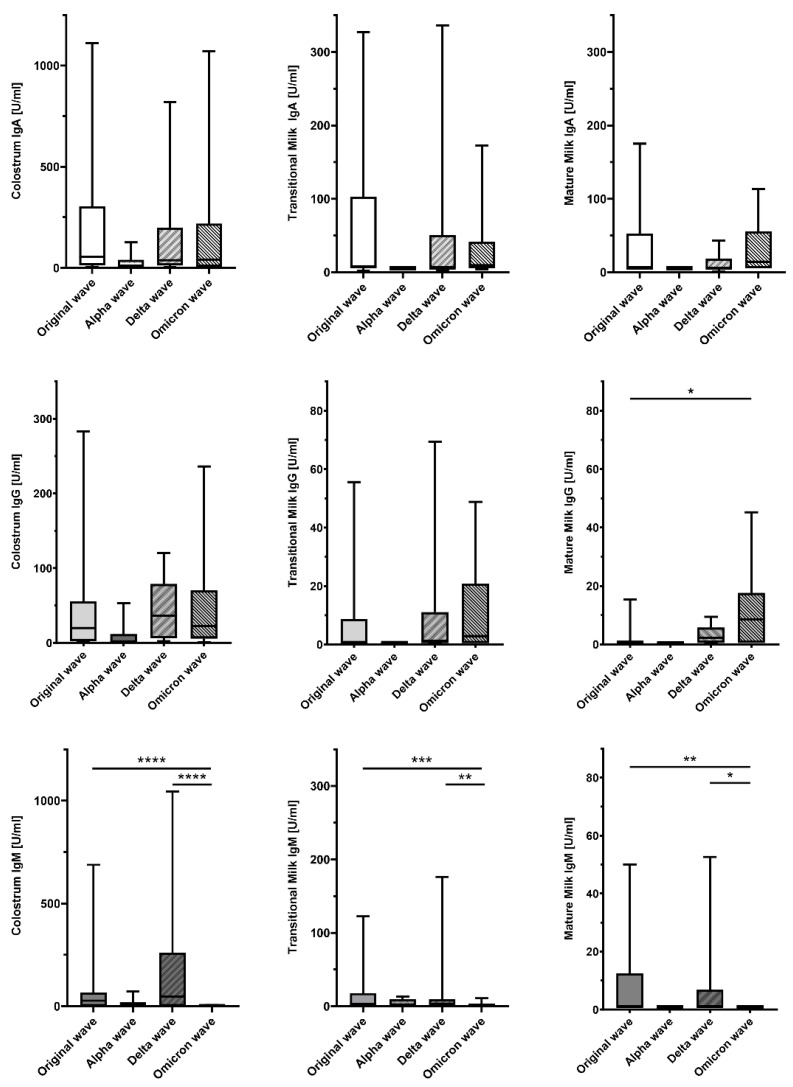
Concentrations of anti-SARS-CoV-2-S1RBD immunoglobulins (Ig) in breast milk across different SARS-CoV-2 waves. This panel displays boxplots of anti-S1RBD-IgA, -IgG, and -IgM concentrations in different types of breast milk (colostrum, transitional milk, and mature milk) across several waves of the SARS-CoV-2 pandemic. Ig concentrations are plotted on the *y*-axis in U/mL. Center lines in boxes represent medians, box edges mark 1st and 3rd quartiles, and whiskers indicate 10th and 90th percentiles. The influence of different SARS-CoV-2 waves on breast milk Ig concentrations is depicted. Significant differences across different pandemic waves are indicated by asterisks (* *p* < 0.05, ** *p* < 0.01, *** *p* < 0.001, **** *p* < 0.0001). *Ig*, immunoglobulin; *SARS-CoV-2*, severe acute respiratory syndrome coronavirus 2; *S1RBD*, SARS-CoV-2 Spike protein S1 receptor-binding domain.

**Table 1 nutrients-16-02320-t001:** Characteristics of study population.

	Total *n* = 140 (100)	Term Birth *n* = 110 (78.6)	Preterm Birth *n* = 30 (21.4)
**Maternal characteristics**			
Maternal age [years]	32.0 ± 4.8	32.0 ± 4.6	31.8 ± 5.6
Gravidity, median (range)	2 (1–3)	2 (1–3)	2 (1–3)
Parity, median (range)	2 (1–2)	2 (1–2)	2 (1–2)
Time of infection, *n* (%)			
during pregnancy	98 (70.0)	82 (74.5)	16 (53.3)
peripartum	42 (30.0)	28 (25.5)	14 (46.7)
Interval infection—delivery [days]	44 (7.5; 102.8)	50 (11; 112.3)	22.5 (1.5; 86.5)
Maternal symptoms, *n* (%)			
a-/pre-symptomatic	22 (15.7)	19 (17.3)	3 (10.0)
mild	95 (67.9)	74 (67.3)	21 (70.0)
moderate	7 (5.0)	5 (4.5)	2 (6.7)
severe	3 (2.1)	0 (0.0)	3 (10.0)
missing information	13 (9.3)	12 (10.9)	1 (3.3)
Vaccination, *n* (%)			
Non-vaccinated	94 (67.1)	72 (65.5)	22 (73.3)
1× vaccinated	5 (3.6)	3 (2.7)	2 (6.7)
2× vaccinated	12 (8.6)	10 (9.1)	2 (6.7)
3× vaccinated	29 (20.7)	25 (22.7)	4 (13.3)
Virus wave, *n* (%)			
Original wave	38 (27.2)	27 (24.5)	11 (36.7)
Alpha wave	9 (6.4)	8 (7.3)	1 (3.3)
Delta wave	29 (20.7)	19 (17.3)	10 (33.3)
Omicron wave	64 (45.7)	56 (50.9)	9 (26.7)
**Neonatal characteristics**			
Number of participants, *n* (%)	144 (100)	111 (77.1)	33 (22.9)
Multiple birth (twins), *n* (%)	4 (2.8)	1 (0.9)	3 (9.1)
Infant sex, *n* (%)			
male	85 (59.0)	65 (58.6)	20 (60.6)
female	59 (41.0)	46 (41.4)	13 (39.4)
Delivery mode			
vaginal	70 (48.6)	62 (55.9)	8 (24.2)
vacuum extraction	6 (4.2)	6 (5.4)	0 (0.0)
C-section	68 (47.2)	43 (38.7)	25 (75.8)
Gestational age [weeks]	38.7 (37.0; 39.8)	39.1 (38.1; 40.1)	34.4 (34.1; 35.3)
Birth weight [g]	3047.3 ± 624.47	3277.4 ± 455.08	2273.2 ± 478.61
z-score birth weight	−0.32 ± 1.00	−0.32 ± 1.00	−0.30 ± 0.83
Birth length [cm]	48.6 ± 3.29	49.7 ± 2.28	44.7 ± 3.30
z-score birth length	−0.72 ± 1.05	−0.70 ± 0.99	−0.81 ± 1.26
Birth head circumference [cm]	34.1 ± 2.21	34.8 ± 1.63	31.8 ± 2.27
z-score birth head circumference	−0.18 ± 1.18	−0.15 ± 1.24	−0.28 ± 0.98
Apgar scores			
1 min	9 (8; 9)	9 (9; 9)	8 (6; 9)
5 min	10 (9; 10)	10 (9.8; 10)	9 (8; 10)
10 min	10 (10; 10)	10 (10; 10)	9 (9; 10)
Umbilical cord arterial pH	7.259 ± 0.073	7.251 ± 0.085	7.293 ± 0.064
Umbilical cord arterial base excess [mmol/L]	−2.7 ± 4.4	−3.0 ± 4.3	−0.9 ± 4.6
Neonate tested positive for SARS-CoV-2, *n* (%)	3 (2.1)	3 (2.7)	0 (0.0)
Neonate admitted to NICU, *n* (%)	34 (23.6)	12 (10.8)	22 (66.7)
DBS umbilical cord blood available, *n* (%)	85 (59.0)	67 (60.3)	18 (54.5)
DBS venous blood neonate 48 h available, *n* (%)	13 (9.0)	10 (9.0)	3 (9.1)
DBS anti-S1RBD-IgG concentrations ^§^	90.8 (23.4; 314.4)	104.3 (27.1; 338.1)	54.6 (15.8; 141.4)
DBS anti-S1RBD-IgM concentrations ^$^	n.d.	n.d.	n.d.

Categorical data are presented as counts (*n*) and percentages (%); continuous data are presented as median (quartile 1; quartile 3) for non-normally distributed variables or mean ± standard deviation (SD) for variables following a normal distribution, unless otherwise specified. ^§^ Limit of detection (LOD) for anti-S1RBD-IgG: 30 U/mL. Data below LOD were censored as follows: LOD/√(2). ^$^ Limit of detection (LOD) for anti-S1RBD-IgM: 100 U/mL. *NICU*, neonatal intensive care unit; *DBS*, dried blood spot(s); *n.d.*, not detectable.

**Table 2 nutrients-16-02320-t002:** Potential factors influencing breast milk immunoglobulin (Ig) concentrations in various sample types.

Type of Breast Milk	Maternal Characteristic	IgA Median (Q1; Q3)	*p*-Value *	IgG Median (Q1; Q3)	*p*-Value *	IgM Median (Q1; Q3)	*p*-Value *
** *Colostrum* **							
	Infection during pregnancy	51.3 (14.0; 221.7)		23.0 (3.7; 60.2)		0.9 (0.9; 51.4)	
	Peripartum infection	23.2 (5.3; 187.9)	0.131	19.3 (4.8; 71.3)	0.896	0.9 (0.9; 3.2)	0.063
	Infection only	24.0 (5.3; 128.5)		11.3 (2.2; 47.2)		7.5 (0.9; 78.7)	
	Infection + vaccination	98.0 (12.0; 457.0)	**0.013**	42.2 (4.0; 112.5)	**0.001**	0.9 (0.9; 0.9)	**<0.001**
	Asymptomatic/mild disease	48.9 (5.7; 214.2)		23.0 (4.8; 66.2)		0.9 (0.9; 47.5)	
	Moderate/severe disease	29.9 (5.3; 1176.0)	0.642	36.3 (9.8; 131.4)	0.206	17.6 (0.9; 475.9)	0.196
	Term birth	38.5 (5.3; 204.8)		20.7 (3.3; 63.9)		0.9 (0.9; 31.2)	
	Preterm birth	36.0 (5.3; 217.7)	0.930	32.9 (10.0; 87.7)	0.236	18.3 (0.9; 183.1)	0.065
	Age < 35 years	40.0 (5.3; 190.2)		22.5 (4.8; 67.7)		0.9 (0.9; 51.4)	
	Age ≥ 35 years	26.0 (5.3; 502.1)	0.619	20.2 (3.1; 59.8)	0.916	0.9 (0.9; 16.2)	0.186
** *Transitional milk* **							
	Infection during pregnancy	5.3 (5.3; 42.5)		0.7 (0.5; 12.2)		0.9 (0.9; 6.6)	
	Peripartum infection	12.5 (5.3; 46.7)	0.308	2.7 (0.5; 21.0)	0.439	0.9 (0.9; 21.2)	0.296
	Infection only	5.3 (5.3; 36.0)		0.5 (0.5; 7.7)		0.9 (0.9; 12.6)	
	Infection + vaccination	20.5 (0.0; 58.6)	0.086	7.3 (0.5; 33.3)	**0.008**	0.9 (0.9; 0.9)	**0.013**
	Asymptomatic/mild disease	5.3 (5.3; 39.2)		0.8 (0.5; 11.4)		0.9 (0.9; 10.6)	
	Moderate/severe disease	46.7 (33.7; 468.5)	**0.012**	20.7 (3.6; 82.4)	**0.026**	12.2 (0.9; 89.5)	0.151
	Term birth	5.3 (5.3; 39.2)		1.4 (0.5; 12.2)		0.9 (0.9; 6.6)	
	Preterm birth	10.6 (5.3; 121.5)	0.325	1.2 (0.5; 44.0)	0.795	0.9 (0.9; 11.9)	0.671
	Age < 35 years	5.3 (5.3; 41.0)		1.2 (0.5; 8.9)		0.9 (0.9; 6.6)	
	Age ≥ 35 years	17.5 (5.3; 102.8)	0.440	3.4 (0.5; 32.3)	0.311	0.9 (0.9; 55.6)	0.333
** *Mature milk* **							
	Infection during pregnancy	5.3 (5.3; 18.4)		0.5 (0.5; 4.6)		0.9 (0.9; 0.9)	
	Peripartum infection	22.5 (8.5; 69.9)	**0.004**	8.5 (2.1; 15.7)	**0.001**	0.9 (0.9; 14.4)	0.248
	Infection only	5.3 (5.3; 18.8)		0.5 (0.5; 2.3)		0.9 (0.9; 5.1)	
	Infection + vaccination	20.4 (4.0; 59.7)	**0.015**	10.0 (5.6; 22.8)	**<0.001**	0.9 (0.9; 0.9)	**0.016**
	Asymptomatic/mild disease	5.3 (5.3; 32.9)		0.6 (0.5; 9.0)		0.9 (0.9; 0.9)	
	Moderate/severe disease	62.0 (21.2; 545.7)	**0.012**	8.2 (3.3; 51.6)	0.136	30.8 (0.9; 69.5)	**0.026**
	Term birth	5.3 (5.3; 32.9)		0.6 (0.5; 9.3)		0.9 (0.9; 0.9)	
	Preterm birth	15.6 (5.3; 59.7)	0.649	2.3 (0.5; 8.5)	0.600	0.9 (0.9; 6.9)	**0.018**
	Age < 35 years	5.3 (5.3; 20.4)		0.5 (0.5; 7.7)		0.9 (0.9; 0.9)	
	Age ≥ 35 years	22.5 (5.3; 69.9)	0.168	5.6 (0.5; 21.6)	0.050	0.9 (0.9; 3.2)	0.722

Data are presented as median (quartile 1 (Q1); quartile 3 (Q3)). * Mann–Whitney-U Test.

**Table 3 nutrients-16-02320-t003:** Breast milk immunoglobulin (Ig) concentrations in various sample types according to virus waves.

Type of Breast Milk	Virus Wave	IgA Median (Q1; Q3)	*p*-Value *	IgG Median (Q1; Q3)	*p*-Value *	IgM Median (Q1; Q3)	*p*-Value *
** *Colostrum* **							
	Original wave	54.6 (13.2; 304.4)		21.5 (2.4; 46.2)		27.2 (0.9; 67.4)	
	Alpha wave	10.7 (1.0; 40.0)		0.9 (0.5; 11.7)		0.9 (0.9; 19.2)	
	Delta wave	38.3 (11.7; 198.7)		36.3 (5.9; 78.8)		47.8 (0.9; 260.8)	
	Omicron wave	41.2 (5.3; 218.7)	0.276	22.2 (5.7; 70.3)	0.075	0.9 (0.9; 0.9)	**<0.001**
** *Transitional milk* **							
	Original wave	8.0 (5.3; 102.8)		0.5 (0.5; 8.8)		3.2 (0.9; 18.1)	
	Alpha wave	5.3 (5.3; 5.3)		0.5 (0.5; 0.5)		0.9 (0.9; 10.0)	
	Delta wave	5.3 (5.3; 50.7)		1.2 (0.5; 11.1)		6.2 (0.9; 45.1)	
	Omicron wave	9.6 (5.3; 41.3)	0.697	2.9 (0.5; 20.8)	0.141	0.9 (0.9; 0.9)	**0.005**
** *Mature milk* **							
	Original wave	5.3 (5.3; 52.5)		0.5 (0.5; 0.9)		0.9 (0.9; 12.4)	
	Alpha wave	5.3 (5.3; 5.3)		0.5 (0.5; 0.6)		0.9 (0.9; 0.9)	
	Delta wave	5.3 (4.3; 18.8)		2.3 (0.5; 5.8)		0.9 (0.9; 6.8)	
	Omicron wave	14.4 (5.3; 55.6)	0.189	8.5 (0.5; 17.6)	**0.004**	0.9 (0.9; 0.9)	**0.004**

Virus waves were categorized according to the timing of infection. Data on the prevailing SARS-CoV-2 variant at any given time point were obtained from the website of the Global Initiative on Sharing all Influenza Data (GISAID). Data are presented as median (quartile 1 (Q1); quartile 3 (Q3)). * Kruskall–Wallis Test.

## Data Availability

All data generated or analyzed during this study are included in this article. Further inquiries can be directed to the corresponding author.

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
