# Peer review of "Factors Influencing Breast Milk Antibody Titers during the Coronavirus Disease 2019 Pandemic: An Observational Study"

_nutrients, 2024, doi:10.3390/nu16142320_

Round 1

Reviewer 1 Report

Comments and Suggestions for Authors

Dear authors,

I have now completed the review of the manuscript titled "Factors Influencing Breast Milk Antibody Titers during the COVID-19 Pandemic: An Observational Study."

In the present study, the authors this study provided valuable data on SARS-CoV-2 antibodies in breast milk.

The manuscript is interesting and, in general, fairly well-written.

I have some suggestions to further improve the quality of the manuscript.

I would like to suggest that the authors address these limitations in the article, either by discussing them in the limitations section or, where feasible, by making the appropriate revisions:

The authors do not consider Associations between Delayed Introduction of Complementary Foods and Childhood Health Consequences in Exclusively Breastfed Children. Discussing this relationship on introduction or discussion will be helpful to the readers. Also, other potential confounding factors like maternal diet, stress levels, etc. were not accounted for.

Lack of viral genome sequencing to definitively determine virus variants. The study relies on timing of infection to infer variant, which may not be fully accurate.

No maternal serum samples were collected, limiting ability to correlate breast milk antibody levels with maternal systemic immune response. Also, if possible, long-term follow-up to assess how antibody levels change over extended periods of lactation would be helpful with comparison to a control group of non-infected/non-vaccinated mothers.

Potential for selection bias, as only mothers giving birth at one hospital in Austria were included. Also, no data on infant outcomes or protection from COVID-19, so clinical significance of antibody levels is unclear. Limited information on other breast milk components beyond antibodies that may contribute to infant immunity. Also it would be better to address the functional capacity of the antibodies detected (e.g. neutralizing ability).

Thank you for your valuable contributions to our field of research. I look forward to receiving the revised manuscript.

Reviewer 2 Report

Comments and Suggestions for Authors

The manuscript titled “Factors Influencing Breast Milk Antibody Titers during the COVID-19 Pandemic: An Observational Study” deals with an interesting topic. The main question addressed by the research is to quantify breast milk Ig specifically directed against the Spike S1 subunit of the SARS-CoV-2 receptor-binding domain (S1RBD) in all stages of lactation throughout the course of the pandemic and to assess factors that influence concentrations of anti-S1RBD Ig in human milk. As soon as the Covid-19 pandemic began, there were doubts about how to manage breastfeeding. This problem has already been addressed in other papers, but the novelty of this manuscript is to elucidate the dynamics of the anti-SARS-CoV-2 antibody response in the different stages of lactation. In addition, given the vaccine hesitancy towards the anti-SARS-CoV-2 vaccine that has been recorded in the general population and in pregnant women residing in some countries of the world, it is important to underline the impact that maternal vaccination status has on antibody titers transmitted to infants.

The article is written in a well-structured manner and the experimental design is appropriate to test the hypothesis. Although the study has limitations, these have been clearly reported by the authors.

The results of the study are interpreted appropriately, and the tables and figures are easy to interpret and understand.

The conclusions are consistent with the evidence and arguments presented and all main questions posed were addressed with specific analyses.

I, also, think that the references are appropriate.

I believe that the paper only needs following changes and clarifications:

2.3.2. Breast milk samples: the abbreviation TMB is present, but the meaning is not specified. Please resolve this inconsistency.

2.4. Statistical analysis: for completeness of the manuscript, I suggest providing a justification of the sample size. Furthermore, the authors report that “Associations between two variables were assessed by means of Pearson correlation coefficient” but I suggest using Spearman’s rank correlation coefficient that is more adequate for your data that are non-normally distributed (such as anti-S1RBD-IgA, -IgM and -IgG in milk and DBS anti-S1RBD-IgG concentrations in neonatal blood samples).

Round 2

Reviewer 1 Report

Comments and Suggestions for Authors

All comments were addressed.